# Emotional vocalizations alter behaviors and neurochemical release into the amygdala

**Zahra Ghasemahmad[1,2,3], Aaron Mrvelj[1], Rishitha Panditi[1], Bhavya Sharma[1], Karthic Drishna Perumal[1]\*, Jeffrey J Wenstrup[1,2,3]\***

[1]Department of Anatomy and Neurobiology and Hearing Research Group, Northeast Ohio Medical University, Rootstown, United States; [2]School of Biomedical Sciences, Kent State University, Kent, United States; [3]Brain Health Research Institute, Kent State University, Kent, United States

**\*For correspondence:**
kdperumal@yahoo.com (KDP);
jjw@neomed.edu (JJW)

**Competing interest:** The authors declare that no competing interests exist.

**Abstract** The basolateral amygdala (BLA), a brain center of emotional expression, contributes to acoustic communication by first interpreting the meaning of social sounds in the context of the listener's internal state, then organizing the appropriate behavioral responses. We propose that modulatory neurochemicals such as acetylcholine (ACh) and dopamine (DA) provide internal-state signals to the BLA while an animal listens to social vocalizations. We tested this in a vocal playback experiment utilizing highly affective vocal sequences associated with either mating or restraint, then sampled and analyzed fluids within the BLA for a broad range of neurochemicals and observed behavioral responses of adult male and female mice. In male mice, playback of restraint vocalizations increased ACh release and usually decreased DA release, while playback of mating sequences evoked the opposite neurochemical release patterns. In non-estrus female mice, patterns of ACh and DA release with mating playback were similar to males. Estrus females, however, showed increased ACh, associated with vigilance, as well as increased DA, associated with reward-seeking. Experimental groups that showed increased ACh release also showed the largest increases in an aversive behavior. These neurochemical release patterns and several behavioral responses depended on a single prior experience with the mating and restraint behaviors. Our results support a model in which ACh and DA provide contextual information to sound analyzing BLA neurons that modulate their output to downstream brain regions controlling behavioral responses to social vocalizations.

## eLife assessment

This **important** study advances our understanding of how distinct types of communication signals differentially affect mouse behaviors and amygdala cholinergic/dopaminergic neuromodulation. The evidence supporting the authors' claims is **solid**. Researchers interested in the complex interaction between prior experience, sex, behavior, hormonal status, and neuromodulation should benefit from this study.

## Introduction

In social interactions utilizing vocal communication signals, the acoustic features of the signals carry emotional information (*Altenmüller et al., 2013*; *Darwin, 1872*; *Seyfarth and Cheney, 2003*). Listeners receive and analyze the acoustic information, compare it with previous experiences, identify the salience and valence of such information, and respond with appropriate behaviors. These

integrated functions depend on brain circuits that include the amygdala, a region located within the temporal lobe that is recognized to play a role in orchestrating emotional responses to salient sensory stimuli (*LeDoux, 2000*; *McGaugh, 2002*; *Sah et al., 2003*). The amygdalar target of auditory inputs from the thalamus and cortex is the basolateral amygdala (BLA) (*LeDoux et al., 1984*; *Romanski and LeDoux, 1993*; *Shi and Cassell, 1997*; *Tsukano et al., 2019*). By integrating this auditory input with other sensory inputs and inputs from other limbic areas, BLA neurons shape appropriate behavioral responses (*Beyeler et al., 2016*; *Gründemann et al., 2019*; *Namburi et al., 2015*; *Namburi et al., 2016*) via projections to downstream targets such as the nucleus accumbens (*Ambroggi et al., 2008*; *Stuber et al., 2011*) and central nucleus of the amygdala (*Ciocchi et al., 2010*).

The BLA processes vocal and other acoustic information in a context-dependent manner (*Gadziola et al., 2016*; *Grimsley et al., 2013*; *Matsumoto et al., 2016*; *Wenstrup et al., 2020*; *Wiethoff et al., 2009*). Contextual information may arise from inputs associated with other sensory modalities (e.g., somatic sensation or olfaction) (*Grimsley et al., 2013*; *Lanuza et al., 2004*; *McDonald, 1998*), but in other cases the contextual information is associated with an animal's internal state. Sources of internal state cues to BLA include brain circuits involving modulatory neurochemicals (i.e., neuromodulators), known to affect the processing of sensory signals, thus shaping attention, emotion, and goal-directed behaviors (*Bargmann, 2012*; *Likhtik and Johansen, 2019*; *Schofield and Hurley, 2018*). While previous work has demonstrated a role for some neuromodulators—dopamine (DA) and acetylcholine (ACh)—in the production of social vocalizations (*Inagaki et al., 2020*, *Rojas-Carvajal et al., 2022*; *Silkstone and Brudzynski, 2020*), it remains unclear how these and other neuromodulators contribute to vocal processing in social interactions.

This study investigates contextual information provided by neuromodulatory inputs to the BLA in response to vocal communication signals. Our hypothesis is that these salient vocalizations elicit distinct patterns of neuromodulator release into the BLA, by which they shape the processing of subsequent meaningful sensory information. We further hypothesize that these neuromodulatory patterns may depend on longer-term processes that are critical to vocal communication: to experience with these behaviors and the accompanying vocalizations, to sex, and to estrous stage in females. To test these hypotheses, we conducted playback experiments in a mouse model to understand the behavioral and neuromodulator responses to salient vocalizations associated with very different behavioral states.

## Results

To study how vocalizations affect behaviors and release of neurochemicals within BLA, we first developed highly salient vocal stimuli associated with appetitive (mating) and aversive (restraint) behaviors of CBA/CaJ mice. From more intense, mating-related interactions between adult male and female mice that included female head-sniffing and attempted or actual mounting (*Gaub et al., 2016*; *Ghasemahmad, 2020*, see Materials and methods), we selected several sequences of vocalizations to form a 20-min mating vocal stimulus. These sequences included ultrasonic vocalizations (USVs) with harmonics, steps, and complex structure, mostly emitted by males, and low-frequency harmonic calls (LFHs) emitted by females (*Figure 1A, C*; *Finton et al., 2017*; *Gaub et al., 2016*; *Ghasemahmad, 2020*; *Hanson and Hurley, 2012*). During short periods of restraint, mice produce distinctive mid-frequency vocalizations (MFVs) that are associated with anxiety-related behaviors and increased release of the stress hormone corticosterone (*Dornellas et al., 2021*; *Grimsley et al., 2016*; *Niemczura et al., 2020*). From vocal sequences produced by restrained mice, we created a 20-min vocal stimulus, primarily containing MFVs and fewer USV and LFH syllables (*Figure 1B, C*).

We next asked whether these salient vocal stimuli, associated with very different behavioral states, could elicit distinct behaviors and patterns of neuromodulator release into the BLA. We focused on the neuromodulators ACh and DA, since previous work suggests that these neuromodulatory systems interact in the emission of positive and negative vocalizations (*Rojas-Carvajal et al., 2022*; *Silkstone and Brudzynski, 2020*). Our experiments combined playback of the vocal stimuli, behavioral tracking and observations (see *Table 1* for descriptions), and microdialysis of BLA extracellular fluid in freely moving mice (*Figure 1D*).

Prior to the study, male and female mouse subjects had no experience with sexual or restraint behaviors. On the first 2 days of the experiment, mice in an experienced group (EXP, *n* = 31) were each exposed to 90-min sessions with mating and restraint behaviors in a counterbalanced design

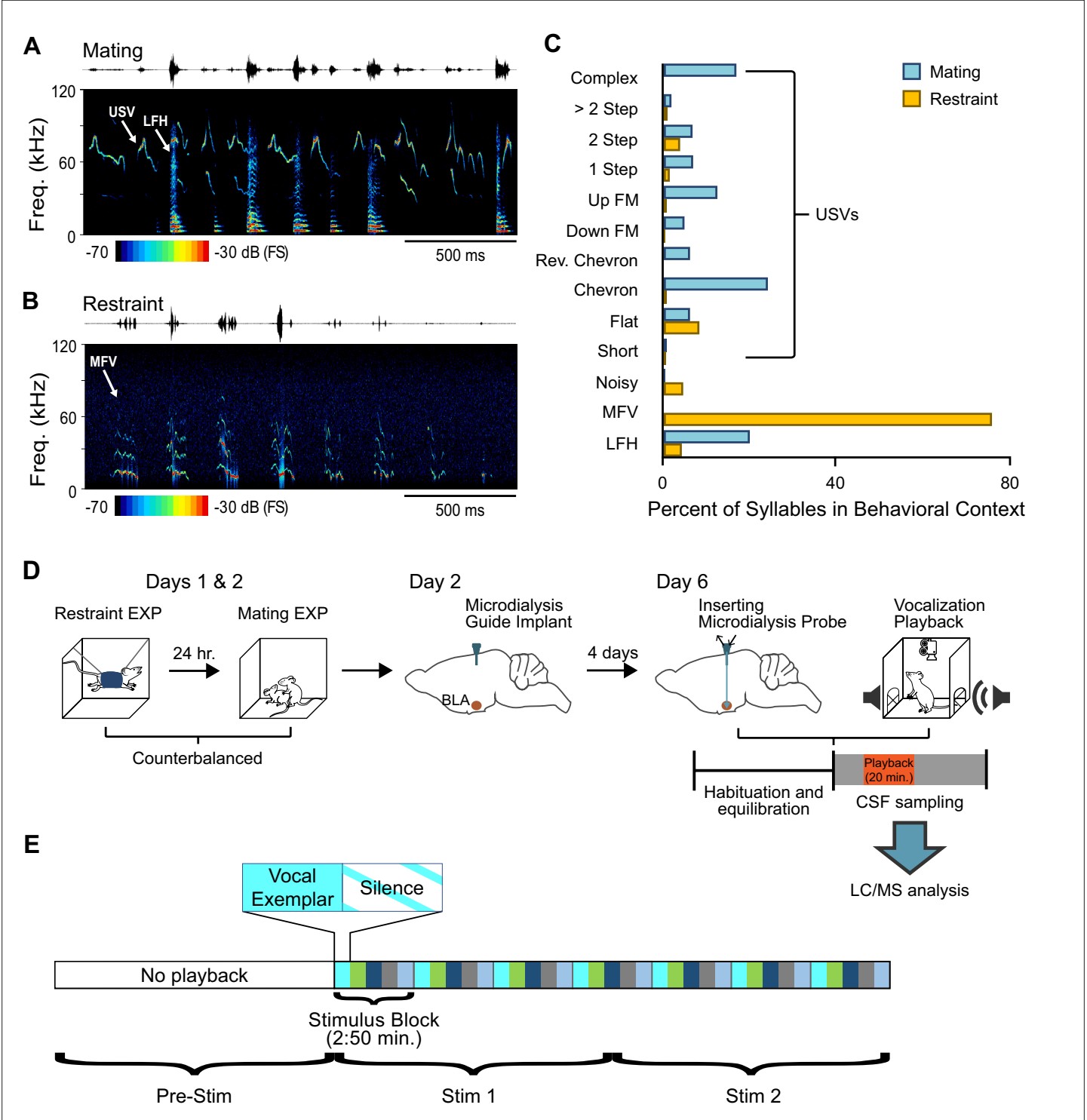

**Figure 1.** Behavioral/microdialysis experiments test how playback of affective vocal signals alters behaviors and neuromodulator release into the basolateral amygdala (BLA). (**A**) Short sample of mating vocal sequence used in playback experiments. Recording was obtained during high-intensity mating interactions and included ultrasonic vocalizations (USVs), likely emitted by the male, as well as low-frequency harmonic (LFH) calls likely emitted by the female. (**B**) Short sample of restraint vocal sequence used in playback experiments. Recording was obtained from an isolated, restrained mouse (see Material and methods) and consisted mostly of mid-frequency vocalization (MFV) syllables. (**C**) Syllable types in mating playback sequences differ substantially from those in restraint playback sequences. Percentages indicate frequency-of-occurrence of syllable types across all examplars used in mating or restraint vocal stimuli ($n_{Mating}$ = 545, $n_{Restraint}$ = 622 syllables). See also *Figure 1—source data 1*. (**D**) Experimental design in playback experiment. Days 1 and 2: each animal experienced restraint and mating behaviors (counterbalanced order across subjects). Day 2: a microdialysis guide

*Figure 1 continued on next page*

*Figure 1 continued*

tube was implanted in the brain above the BLA. Day 6: the microdialysis probe was inserted through guide tube into the BLA. Playback experiments began after several hours of habituation/equilibration. Behavioral observations and microdialysis sampling were obtained before, during, and after playback of one vocalization type. Microdialysis samples were analyzed using liquid chromatography/mass spectrometry (LC/MS) method described in Materials and methods. (**E**) Schematic illustration of detailed sequencing of vocal stimuli, shown here for mating playback. A 20-min period of vocal playback was formed by seven repeated stimulus blocks of 170 s. The stimulus blocks were composed of five vocal exemplars (each represented by a different color) of variable length, with each exemplar followed by an equal duration of silence. Stimuli during the Stim 1 and Stim 2 playback windows thus included identical blocks but in slightly altered patterns. See Material and methods for in-depth description of vocalization playback.

The online version of this article includes the following source data for figure 1:

**Source data 1.** This source data file identifies each syllable occurrence throughout the vocal examplars used in mating and restraint playback and summarized in *Figure 1C*.

(*Figure 1D*). Mice were then implanted with a guide cannula for microdialysis. On the playback/sample collection day (Day 6), a microdialysis probe was inserted into the guide cannula. After a 4-hr period of mouse habituation and probe equilibration, we recorded behavioral reactions and sampled extracellular fluid from the BLA before (Pre-Stim) and during a 20-min playback period, divided into two 10-min stimulation/collection/observation periods that are designated Stim 1 and Stim 2 (*Figure 1D, E*; see also Materials and methods). Each mouse received playback of either the mating or restraint stimuli, but not both: same-day presentation of both stimuli would require excessively long playback sessions, the condition of the same probe would likely change on subsequent days, and quality of a second implanted probe on a subsequent day was uncertain. Fluids were analyzed using a liquid chromatography/mass spectrometry (LC/MS) technique that allowed simultaneous measurement of several neurochemicals and their metabolites in the same dialysate samples, including ACh, DA, and the serotonin metabolite 5-hydroxyindoleacetic acid (5-HIAA) (see Materials and methods). All neurochemical results during Stim 1 and Stim 2 periods are expressed as a percentage relative to the Pre-Stim control period. However, raw values of ACh, DA, and 5-HIAA are reported in source data files (*Figure 3—source data 1*, *Figure 3—source data 2*, *Figure 3—source data 3*). Data are reported only from mice with more than 75% of the microdialysis probe implanted within the BLA (*Figure 2*).

We first describe tests to examine whether playback of mating and restraint vocalizations results in different behavioral and neurochemical responses in male mice. We observed that two behaviors, still-and-alert and flinching, showed differential effects of playback type, increasing during restraint playback relative to mating playback during the Stim 1 period (*Figure 3A, B*). There were also distinct patterns of ACh and DA release into the BLA depending on the type of vocalization playback (*Figure 3C, D*). Thus, in response to restraint vocalizations, we observed an increase in

**Table 1.** Classification of manually evaluated behaviors during playback of vocalizations.

| Behavior | Definition |
| --- | --- |
| Abrupt attending | Sudden and quick pause in locomotion followed by abrupt change in head and body position lasting 2+ s. This behavior is accompanied by fixation of the eyes and ears during attending. Appears similar to 'freezing' behavior described in fear conditioning but occurs in the context of natural response to vocalizations. |
| Flinch | A short-duration twitch-like movement in response to vocal sequences, occurring at any location within arena. Unlike acoustic startle (*Grimsley et al., 2015*), this behavior occurs in free-moving animals in response to non-repetitive, variable-level vocal stimuli. Flinching movements are of smaller magnitude than those observed in acoustic startle. |
| Locomotion | Movement of all four limbs from one quadrant of the arena to another. |
| Rearing | A search behavior during which the body is upright and the head is elevated to investigate more distant locations. |
| Self-grooming | Licking fur and using forepaws to scratch and clean fur on head. The behavior can extend to other parts of the body. |
| Still-and-alert | Gradual reduction or lack of movement for 2+ s, during which animal appears to be listening or responding to external stimulus. Distinguished from abrupt attending by gradual onset. |
| Stretch-attend posture | A risk-assessment behavior during which hind limbs are fixed while the head, forelimbs, and the body are stretched sequentially in different directions. |

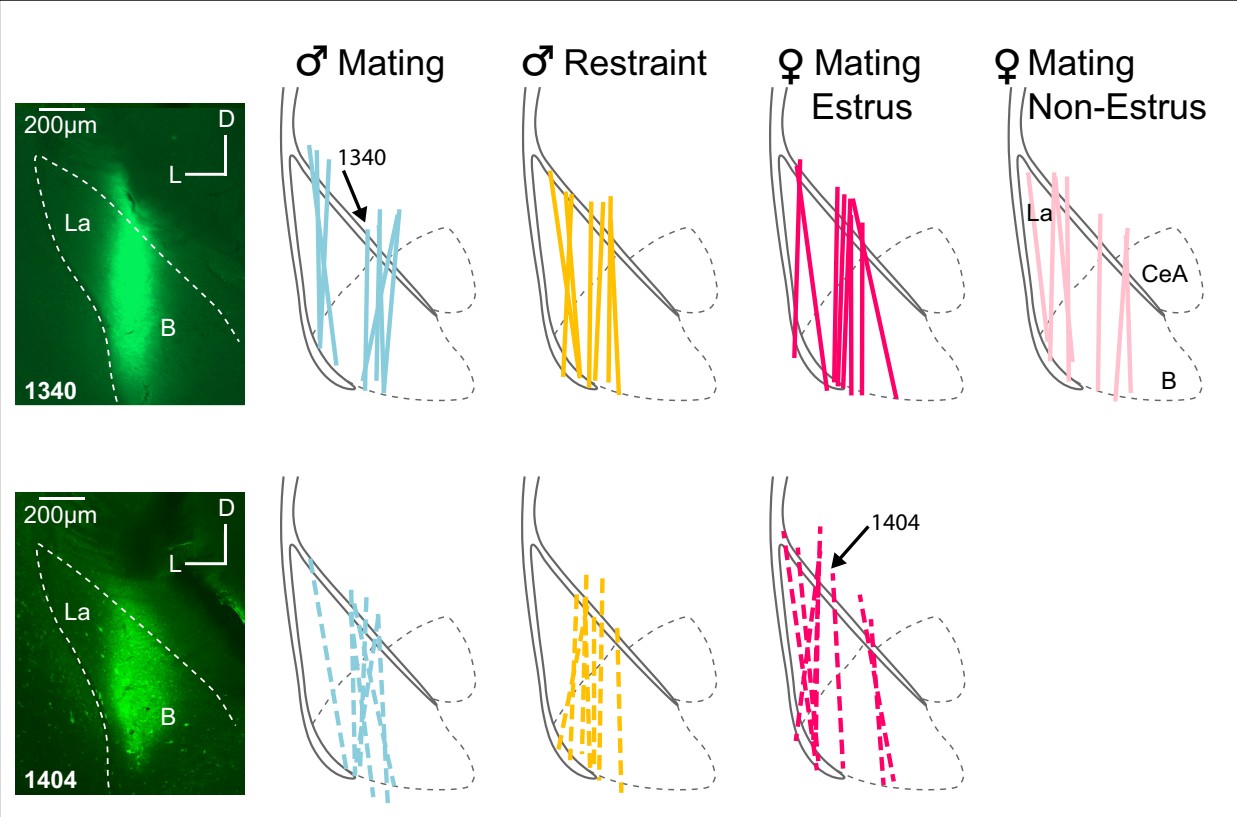

**Figure 2.** Microdialysis probe locations for EXP and INEXP groups. Labels above basolateral amygdala (BLA) outlines indicate groups based on playback type, sex, and hormonal state. Colored lines indicate recovered probe tracks that resulted from infusion of fluorescent tracers. Black solid lines indicate external capsule; black dashed lines indicate major amygdalar subdivisions. Arrows indicate tracks related to insets. Insets: photomicrographs show dextran-fluorescein labeling that marks the placement of the microdialysis probes for two cases, 1340 and 1404. Abbreviations: *B*, basal nucleus of amygdala; *CeA*, central nucleus of amygdala; *La*, lateral nucleus of amygdala.

ACh concentration during both Stim 1 (*M* ± *SD*: +24 ± 31% re Pre-Stim baseline) and Stim 2 (+25 ± 16%) playback windows. Mating vocalizations, however, resulted in a decrease in ACh release during both playback periods (Stim 1: −19 ± 21.0%; Stim 2: −16.4 ± 24.0%) (*Figure 3C*). DA release displayed opposite patterns to ACh, increasing during playback of mating vocalizations during Stim 1 (+12.0 ± 27.4%) and Stim 2 (+25 ± 39% re baseline), but decreasing during playback of restraint vocal sequences (Stim 1: −11.0 ± 11.2%; Stim 2: −22 ± 25.6%) (*Figure 3D*). For ACh, levels differed significantly between the mating and restraint groups during both Stim 1 and Stim 2, while DA levels differed significantly only during Stim 2 (*Figure 3C, D*). In contrast, the serotonin metabolite 5-HIAA showed no distinct pattern over time following playback of either vocal stimulus, nor significant differences between groups (*Figure 3E*). These findings suggest that both behavioral responses and ACh and DA release are modulated in listening male mice by the affective content of social vocalizations.

As male and female mice emit different vocalizations during courtship and mating (*Finton et al., 2017*; *Grimsley et al., 2013*; *Neunuebel et al., 2015*; *Sales, 1972*), we tested whether playback of vocal interactions associated with mating (*Figure 1A*) results in different behavioral and neurochemical responses in listening male and female mice. Since our testing included both estrus and non-estrus females, we further examined the estrous effect on neurochemical release and behavioral reactions.

Playback of mating vocalizations resulted in some general and some sex-based differences in behavioral responses. For instance, all groups displayed increased attending behavior (*Figure 4A*). In females, regardless of estrous stage, rearing decreased and still-and-alert behavior increased significantly relative to male mice (*Figure 4B, C*). This change in motor activity was further supported by video tracking results showing a significant decrease in distance traveled by female mice in response to mating vocal playback (significant sex effect ($F(1,15) = 9.3$, p = 0.008, $\eta^2 = 0.4$)). However, one behavioral change was estrous dependent: females in estrus displayed a strikingly higher number of

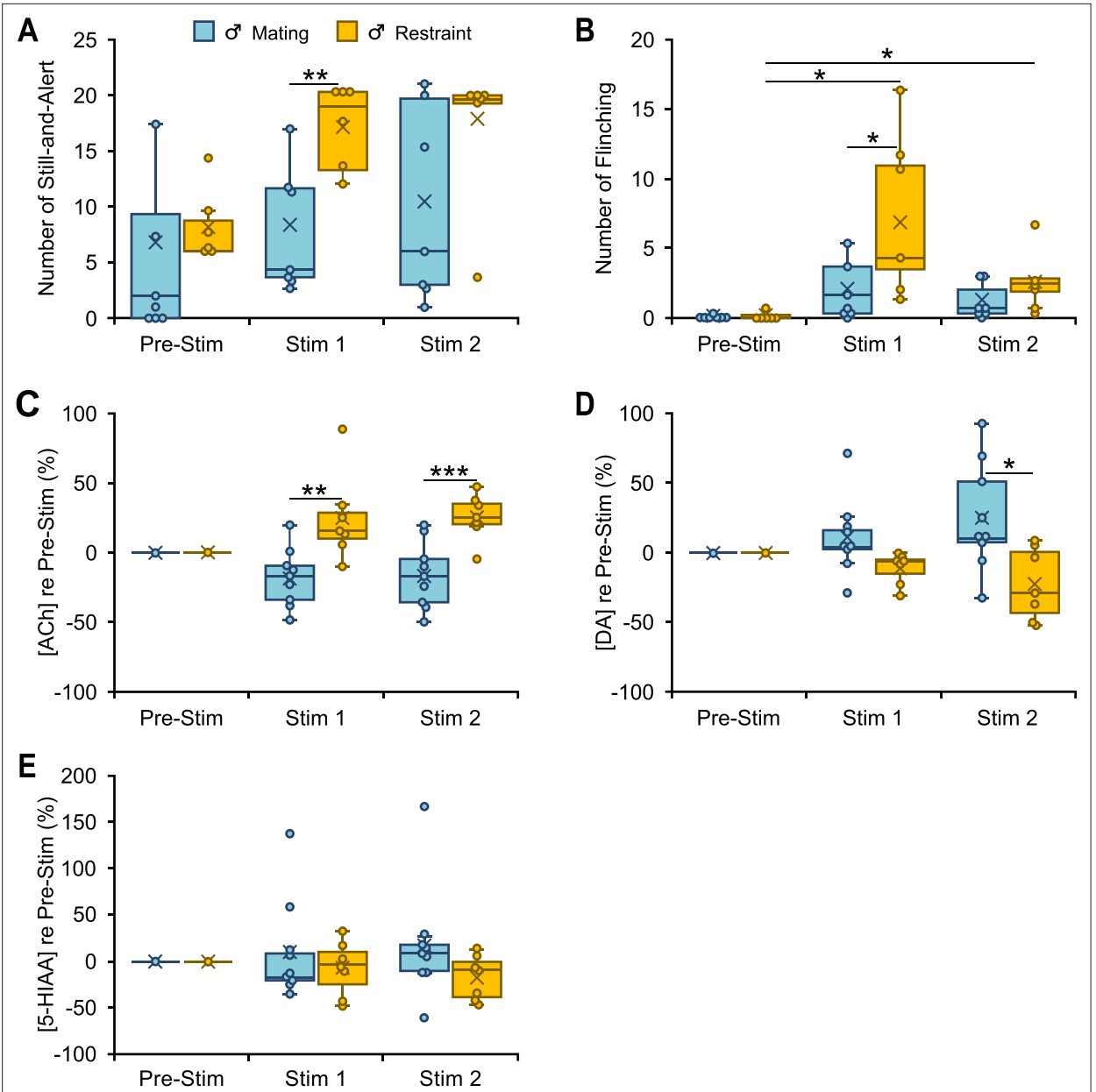

**Figure 3.** Behavioral and neuromodulator responses to vocal playback in male mice differ by behavioral context of vocalizations. (**A, B**) Boxplots show number of occurrences of specified behavior in 10-min observation periods before (Pre-Stim) and during (Stim 1, Stim 2) playback of mating or restraint vocal sequences ($n_{Mating} = 7$, $n_{Restraint} = 6$). Note that playback sequences during Stim 1 and Stim 2 periods were identical within each group. During playback, the restraint group increased still-and-alert behavior compared to the mating group (**A**) (context: $F(1,11) = 9.6$, p = 0.01, $\eta^2 = 0.5$), and increased flinching behavior (**B**) (time*context: $F(1.1,12.2) = 6.3$, p = 0.02, $\eta^2 = 0.4$) compared to the Pre-Stim baseline and to the mating group. (**C–E**) Boxplots show differential release of acetylcholine (ACh), dopamine (DA), and 5-hydroxyindoleacetic acid (5-HIAA) relative to the Pre-Stim period, during mating and restraint vocal playback ($n_{Mating} = 9$, $n_{Restraint} = 7$). (**C**) Significant differences in ACh for restraint (increase) playback in Stim 1 and Stim 2 vs males in mating playback (Main effect of context: $F(1,14) = 22.6$, p < 0.001, $\eta^2 = 0.62$). (**D**) Significant differences for DA for mating (increase) vs restraint playback (Main effect of context: $F(1,14) = 7.4$, p = 0.02, $\eta^2 = 0.35$). (**E**) No significant changes in 5-HIAA during vocal playback in male mice (context: $F(1,14) = 1.36$, p = 0.3, $\eta^2 = 0.09$). (**A–E**) Statistical testing examined time windows within groups for behavioral observations and all intergroup comparisons within time windows for both behavior and neuromodulator data. Only normalized neuromodulators in Stim 1 and Stim 2 were used for statistical comparisons. Only significant tests are shown. Repeated measures generalized linear model (GLM): *p < 0.05, **p < 0.01, ***p < 0.001 (Bonferroni post hoc test). Time windows comparison: 95% confidence intervals. See Data analysis section in *Materials and methods* for description of box plots. See *Figure 3—source data 1–4* for all numerical data for *Figures 3–6*.

The online version of this article includes the following source data for figure 3:

*Figure 3 continued on next page*

*Figure 3 continued*

**Source data 1.** This source data file shows the raw and normalized values of acetylcholine (ACh) concentration for each measurement displayed in *Figures 3–6*.

**Source data 2.** This source data file shows the raw and normalized values of dopamine (DA) concentration for each measurement displayed in *Figures 3–6*.

**Source data 3.** This source data file shows the raw and normalized values of 5-HIAA concentration for each measurement displayed in *Figures 3, 4 and 6*.

**Source data 4.** This source data file shows the values for behavioral events and tracking data for each measurement displayed in *Figures 3–5*.

flinching behaviors compared to males and non-estrus females during both Stim 1 and Stim 2 periods (*Figure 4D*). Our analysis of neuromodulator responses to mating vocalization playback revealed an estrous-dependent modulation of ACh levels during playback. ACh concentration in estrus females increased during both Stim 1 and Stim 2 periods, whereas ACh decreased in both males and non-estrus females (*Figure 4E*). Moreover, during playback, post hoc comparison with Bonferroni correction showed that the ACh in estrus females was significantly higher (Stim 1, +29.3 ± 27.0%; Stim 2, +25.5 ± 22%) than both males (Stim 1, −18 ± 21.0%; Stim 2, −16.0 ± 24.0%) and non-estrus females (Stim 1, −6.0 ± 14%; Stim 2, −35.0 ± 29.6%). DA release showed increases during mating playback within all three experimental groups (males: Stim 1, +12.0 ± 27.0%, Stim 2, +26.0 ± 39.1%; estrus females: Stim 1, +24.1 ± 37.0%, Stim 2, +23.0 ± 27.0%; non-estrus females: Stim 1, +27.0 ± 17.0%, Stim 2, +49.0 ± 48.0%), but no significant differences among groups (*Figure 4F*). Similar to male groups in restraint and mating playback, the 5-HIAA release patterns in females showed no clear modulation pattern during mating vocal playback or differences among groups (*Figure 4G*).

Like male mice exposed to restraint vocalizations, estrus females showed robust and significant increases in flinching behavior in response to mating vocalizations. Both groups displayed increased ACh release. This supports the possible involvement of ACh in shaping such behavior in both males listening to restraint calls and in estrus females listening to mating vocalizations. Note that neuromodulator release, including ACh, has been previously linked to motor behaviors (*Wall and Woolley, 2020*), but the changes in ACh that we observed showed no relationship with behaviors involving motor activity such as rearing (restraint males: Stim 1: $n = 6$, $r = 0.1$, $p = 0.8$; Est females: $n = 6$, $r = 0.7$, $p = 0.08$), locomotion (Stim 1: restraint males, $n = 6$, $r = −0.2$, $p = 0.7$; Est females: $n = 6$, $r = 0.3$, $p = 0.6$), or distance traveled (Stim 1: restraint males, $n = 6$, $r = 0.16$, $p = 0.8$; Est females: $n = 6$, $r = −0.6$, $p = 0.2$). This suggests that the observed changes in ACh reflect the valence of these vocalizations.

All EXP mice used in the above experiments had undergone a single session each to experience mating and restraint conditions prior to the playback session on Day 6 (*Figure 1D*). Does such experience shape the release patterns of these neuromodulators in response to vocal playback? We tested male and female mice under identical vocal playback conditions as previous groups, except that they did not receive the restraint and mating experiences (INEXP groups). Since only one INEXP female was in a non-estrus stage during the playback session, our analysis of the effect of experience included only estrus females and males.

Several behavioral responses to vocalization playback differed between EXP and INEXP mice in a sex- or context-dependent manner. For example, EXP estrus females showed significantly increased sound-evoked flinching behaviors compared to INEXP estrus females (*Figure 5A*) in response to mating vocal sequences. These experience effects were not observed in males in response to mating or restraint vocal playback. Furthermore, the differences in flinching behavior among EXP groups (male-mating vs male-restraint or estrus female-mating) were not evident among INEXP groups. For rearing behavior, EXP males responding to mating vocalizations showed an increase compared to INEXP males (*Figure 5B*). This pattern was not observed during restraint playback for males or mating playback in estrus females. These findings indicate that behavioral responses to salient vocalizations result from interactions between sex of the listener or context of vocal stimuli with the previous behavioral experience associated with these vocalizations.

A major finding is that prior experience with mating and restraint behaviors shaped patterns of ACh release in response to vocal playback; there was a significant interaction of context and experience for ACh ($F(1,39) = 12.7$, $p < 0.001$; *Figures 5C and 6A*). Thus, ACh release was significantly different between INEXP and EXP males in restraint playback groups. A similar pattern was observed in INEXP estrus females in response to mating vocal playback, that is, the concentration of ACh re baseline

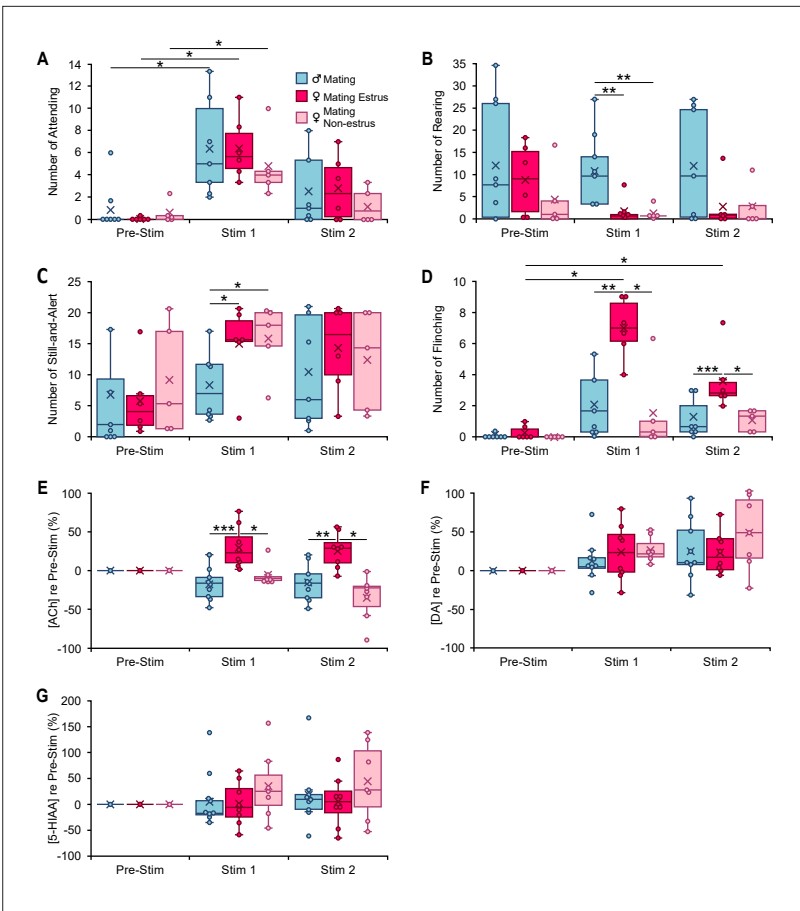

**Figure 4.** Behavioral and neuromodulator responses to playback of mating vocal sequences differ by sex and female estrous stage. (**A–D**). Occurrences of specified behaviors in 10-min periods before (Pre-Stim) and during (Stim 1, Stim 2) mating vocal playback ($n_{Male}$ = 7, $n_{Estrus\ Fem}$ = 6, $n_{Non-estrus\ Fem}$ = 5). (**A**) Attending behavior increased during Stim 1 in response to mating vocal playback, regardless of sex or estrous stage (time: $F(2,30)$ = 32.6, p < 0.001, $\eta^2$ = 0.7; time*sex: $F(2,30)$ = 0.12; p = 0.9, $\eta^2$ = 0.008; time*estrous: $F(2,30)$ = 1.1; p = 0.4, $\eta^2$ = 0.07). (**B, C**) Females regardless of estrous stage reared less (sex: $F(1,15)$ = 10.22, p = 0.006; $\eta^2$ = 0.4; estrous: $F(1,15)$ = 0.2, p = 0.7, $\eta^2$ = 0.01) and displayed more Still-and-Alert behaviors (sex: $F(1,15)$ = 5.2, p = 0.04, partial $\eta^2$ = 0.3, estrous: $F(1,15)$ = 0.07, p = 0.8, partial $\eta^2$ = 0.005) than males during mating vocal playback. (**D**) Estrus females, but not non-estrus females or males, showed a significant increase in flinching behavior during Stim 1 and Stim 2 periods (time*estrous: $F(2,30)$ = 9.0, p < 0.001, $\eta^2$ = 0.4). (**E–G**) Changes in concentration of acetylcholine (ACh), dopamine (DA), and 5-hydroxyindoleacetic acid (5-HIAA) relative to the Pre-Stim period, evoked during Stim 1 and Stim 2 periods of vocal playback ($n_{Male}$ = 9, $n_{Estrus\ Fem}$ = 8, $n_{Non-estrus\ Fem}$ = 7). (**E**) Release of ACh during mating playback increased in estrus females (Stim 1, Stim 2) but decreased in males and non-estrus females (Stim 2). Among groups, there was a significant estrous effect (estrous: $F(1,21)$ = 29.0, p < 0.001, $\eta^2$ = 0.6). (**F**) DA release during mating playback increased in all groups relative to Pre-Stim period, with no significant sex ($F(1,21)$ = 0.8, p = 0.4, $\eta^2$ = 0.04) or estrous effect ($F(1,21)$ = 0.9, p = 0.4, $\eta^2$ = 0.04). (**G**) No significant changes in 5-HIAA during mating sequence playback (sex: $F(1,21)$ = 0.07, p = 0.8, $\eta^2$ = 0.004; estrus: $F(1,21)$ = 1.6, p = 0.22, $\eta^2$ = 0.07). (**A–G**) Repeated measures generalized linear model (GLM): *p < 0.05, **p < 0.01, ***p < 0.001 (Bonferroni post hoc test). Time windows comparison: 95% confidence intervals.

failed to show such pronounced increases as observed in EXP females and showed a significant experience effect (sex*experience interaction: $F(1,39)$ = 7.8, p = 0.008; **Figures 5C and 6A**). Furthermore, INEXP males in mating playback group failed to show the reduced ACh concentration observed in EXP males (INEXP Stim 1: +9.6 ± 15.0%; INEXP Stim 2: +6.1 ± 30.0; EXP Stim 1: −19.0 ± 21.0%; EXP Stim 2: −16.0 ± 24.0%), even though these differences were not significant (**Figures 5C and 6A**).

DA release patterns, which were weaker in EXP groups compared to ACh patterns (**Figure 3C vs D** ; **Figure 4E vs F**) showed no significant experience-dependent changes in any group (**Figure 5D**).

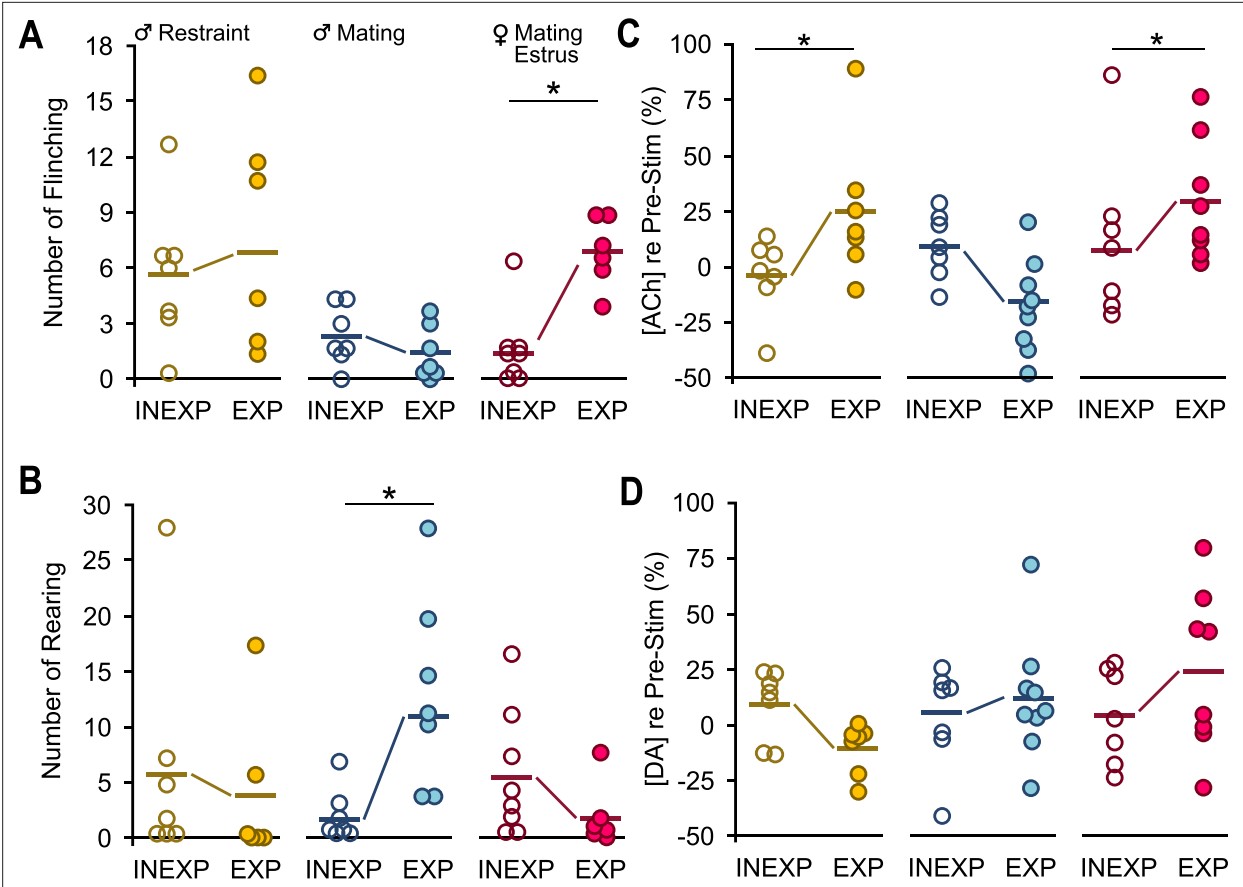

**Figure 5.** A single bout each of mating and restraint experience altered behaviors and acetylcholine (ACh) release in response to vocal playback. In all graphs, dots represent measures from individual animals obtained during the Stim 1 playback period; thick horizontal lines represent mean values across subjects. There were no differences in behavioral counts for Pre-Stim values between INEXP and EXP mice for any group (e.g., male-restraint, male-mating, estrus female-mating). Neuromodulator values are normalized to the baseline level. (**A**) Experience increased flinching responses in estrus female mice but not in males in mating or restraint vocal playback (time*sex*experience: $F(1.6,56) = 4.1$, $p = 0.03$, $\eta^2 = 0.11$). (**B**) Experience increased rearing responses in males exposed to mating playback (sex*experience: $F(1,35) = 5.3$, $p = 0.03$, $\eta^2 = 0.13$). (**C**) Estrus female mice (sex*experience: $F(1,39) = 8.0$, $p = 0.008$, $\eta^2 = 0.2$) and restraint male mice (context*experience $= F(1,39) = 13.0$, $p < 0.001$, $\eta^2 = 0.2$) displayed consistent experience effect for changes in ACh. (**D**) Dopamine (DA) did not show the EXP effect observed in ACh during vocal playback (sex*experience: $F(1,39) = 0.12$, $p = 0.7$, $\eta^2 = 0.003$; context*experience: $F(1,39) = 4.0$, $p = 0.052$, $\eta^2 = 0.09$). Generalized linear model (GLM) repeated measures with Bonferroni post hoc test: *$p < 0.05$.

There was no evidence that vocal playback altered DA release in any of the three groups of INEXP mice (*Figure 6B*), unlike what was observed in EXP groups (*Figures 3D and 4F*). 5-HIAA concentrations, which were unaffected by sex, estrous stage, or playback type (*Figures 3E and 4G*), were also unaffected by experience (*Figure 6C*).

Collectively, these data suggest that the playback vocalization type and estrous effects observed in ACh release patterns and behavioral reactions depend on previous experience with the corresponding behaviors.

## Discussion

Functional imaging studies in humans and mechanistic studies in other species provide substantial evidence that the amygdala participates in circuits that process vocalizations (*Frühholz et al., 2016*; *Liebenthal et al., 2016*; *Sander et al., 2003*; *Wenstrup et al., 2020*; *Voytenko et al., 2023*), assess the valence of appetitive and aversive cues (*Kyriazi et al., 2018*; *O'Neill et al., 2018*; *Pignatelli and Beyeler, 2019*; *Smith and Torregrossa, 2021*), and shape appropriate behavioral responses to these cues (*Gründemann et al., 2019*; *Lim et al., 2009*; *Schönfeld et al., 2020*; *Zhang and Li,*

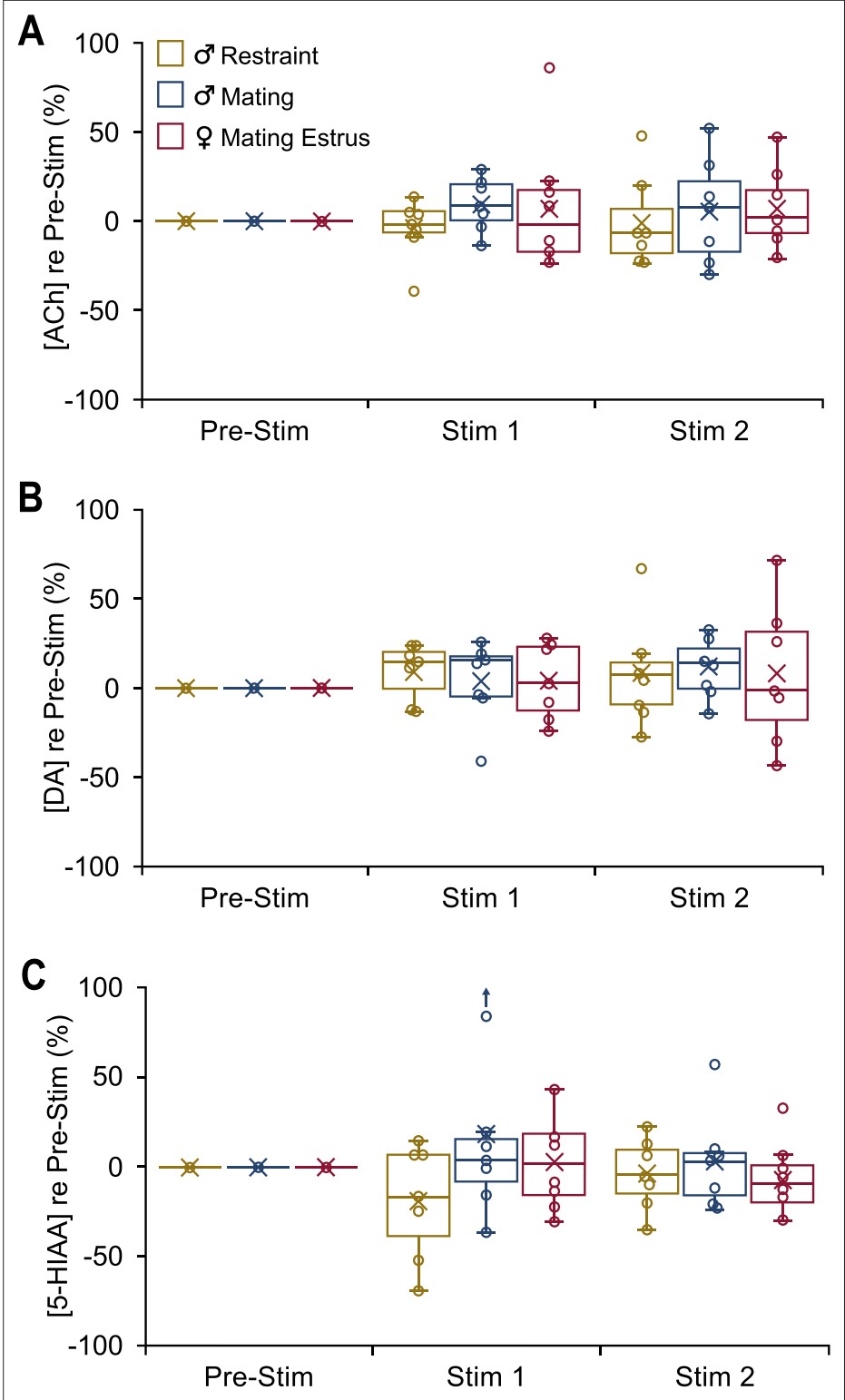

**Figure 6.** In INEXP mice, vocal playback failed to evoke distinct patterns in neuromodulator release. Boxplots show change in concentration of the specified neuromodulator in 10-min playback periods (Stim 1, Stim 2) compared to the baseline level (Pre-Stim). Male-restraint, $n_{INEXP}$ = 7; male-mating, $n_{INEXP}$ = 7; estrus female-mating, $n_{INEXP}$ = 7. (**A**) Acetylcholine (ACh) release shown for both playback periods. (**B**) Dopamine (DA) release shown for both playback periods. (**C**) 5-Hydroxyindoleacetic acid (5-HIAA) concentration shown for both playback periods.

*2018*). Nonetheless, how contextual information is delivered to the amygdala and contributes to vocal processing is not well understood. Since the amygdala receives strong projections from neuro-modulatory brain centers (*Aitta-Aho et al., 2018*; *Asan, 1997*; *Asan, 1998*; *Bigos et al., 2008*; *Bocchio et al., 2016*; *Carlsen et al., 1985*), and since the role of these neurochemicals in providing internal state and contextual information is well documented (*Bocchio et al., 2016*; *Jiang et al., 2016*; *Likhtik and Johansen, 2019*), we hypothesized that release patterns of neuromodulators into the BLA provide contextual information during processing of affective vocalizations. Our results show that these emotionally charged vocalizations result in distinct release patterns of ACh and DA into the BLA of male and female mice. Furthermore, female hormonal state appears to influence ACh but not DA release into the BLA when processing mating vocalizations. Such context- or state-dependent changes were not observed in patterns of other neurochemicals (e.g., 5-HIAA), nor in the absence of experience with behaviors associated with the vocalizations. In particular, we showed that a single 90-min experience with intense behaviors is sufficient to establish strong, consistent patterns of ACh release into the amygdala. These data indicate that during analysis of affective vocalizations in the BLA, ACh, and DA provide experience-dependent, state- and context-related information that can potentially modulate sensory processing within the BLA and thus shape an individual's response to these vocalizations.

The linkage between neuromodulator release and behavioral responses to vocalizations varies. The strongest case is for a link between increased ACh release and flinching behavior. Males in the restraint playback group and estrus females in the mating playback group both displayed significantly increased ACh release and flinching behavior, occurring in both playback periods. This suggests a mechanistic relationship. Other significant behavioral responses to vocal stimuli did not closely match the timing of neurochemical changes, suggesting a weak mechanistic link. Overall, it is not surprising that most behaviors and neurochemicals do not match well, given the poor temporal resolution of the neurochemical sampling. We believe that our results show the need for a finer analysis of DA and ACh release based on techniques for high temporal resolution of neurochemical measurements (e.g., genetically encoded ACh indicator; *Jing et al., 2018*), coupled with short interval behavioral observations. Only then can the role of the BLA and neuromodulator inputs in behavioral vocal responses be established more strongly. Furthermore, we note that no conclusion can be made regarding relationships between noradrenalin and serotonin concentrations and vocal playback, since these were not detected by the neurochemical analysis.

The BLA receives strong cholinergic projections from the basal forebrain (*Aitta-Aho et al., 2018*; *Carlsen et al., 1985*) that contribute to ACh-dependent processing of aversive cues and fear learning in the amygdala (*Baysinger et al., 2012*; *Gorka et al., 2015*; *Jiang et al., 2016*; *Tingley et al., 2014*). Our findings support these studies by demonstrating increased ACh release in BLA in EXP animals in response to playback of aversive vocalizations. Although the exact mechanism by which ACh affects vocal information processing in BLA is not clear yet, the result of ACh release onto BLA neurons seems to enhance arousal during emotional processing (*Likhtik and Johansen, 2019*). Our results, in conjunction with previous work, suggest mechanisms by which vocalizations affect ACh release and in turn drive behavioral responses (*Figure 7A*). Cholinergic modulation in the BLA is mediated via muscarinic and nicotinic ACh receptors on BLA pyramidal neurons and inhibitory interneurons (*Aitta-Aho et al., 2018*; *Mesulam et al., 1983*; *Pidoplichko et al., 2013*; *Unal et al., 2015*). During the processing of sensory information in the BLA, partially non-overlapping populations of neurons respond to cues related to positive or negative experiences (*Namburi et al., 2015*; *Paton et al., 2006*; *Smith and Torregrossa, 2021*). These neurons then project to different target areas involved in appetitive or aversive behaviors—the nucleus accumbens or central nucleus of the amygdala, respectively (*Namburi et al., 2015*).

In response to an aversive cue or experience (*Figure 7A*), released ACh affects BLA neurons according to their activity. If projection neurons are at rest, ACh may exert an inhibitory effect by activating nicotinic ACh receptors on local GABAergic interneurons, which in turn synapse onto the quiescent pyramidal neurons. This results in GABA-A-mediated inhibitory postsynaptic potentials in the pyramidal neurons. Alternately, direct activation of M1 ACh receptors on pyramidal neurons, activating inward rectifying $K^+$ currents, may result in additional ACh-mediated inhibition (*Figure 7A*; *Aitta-Aho et al., 2018*; *Pidoplichko et al., 2013*; *Unal et al., 2015*). When BLA pyramidal neurons are already active due to strong excitatory input associated with aversive cues, M1 receptor activation

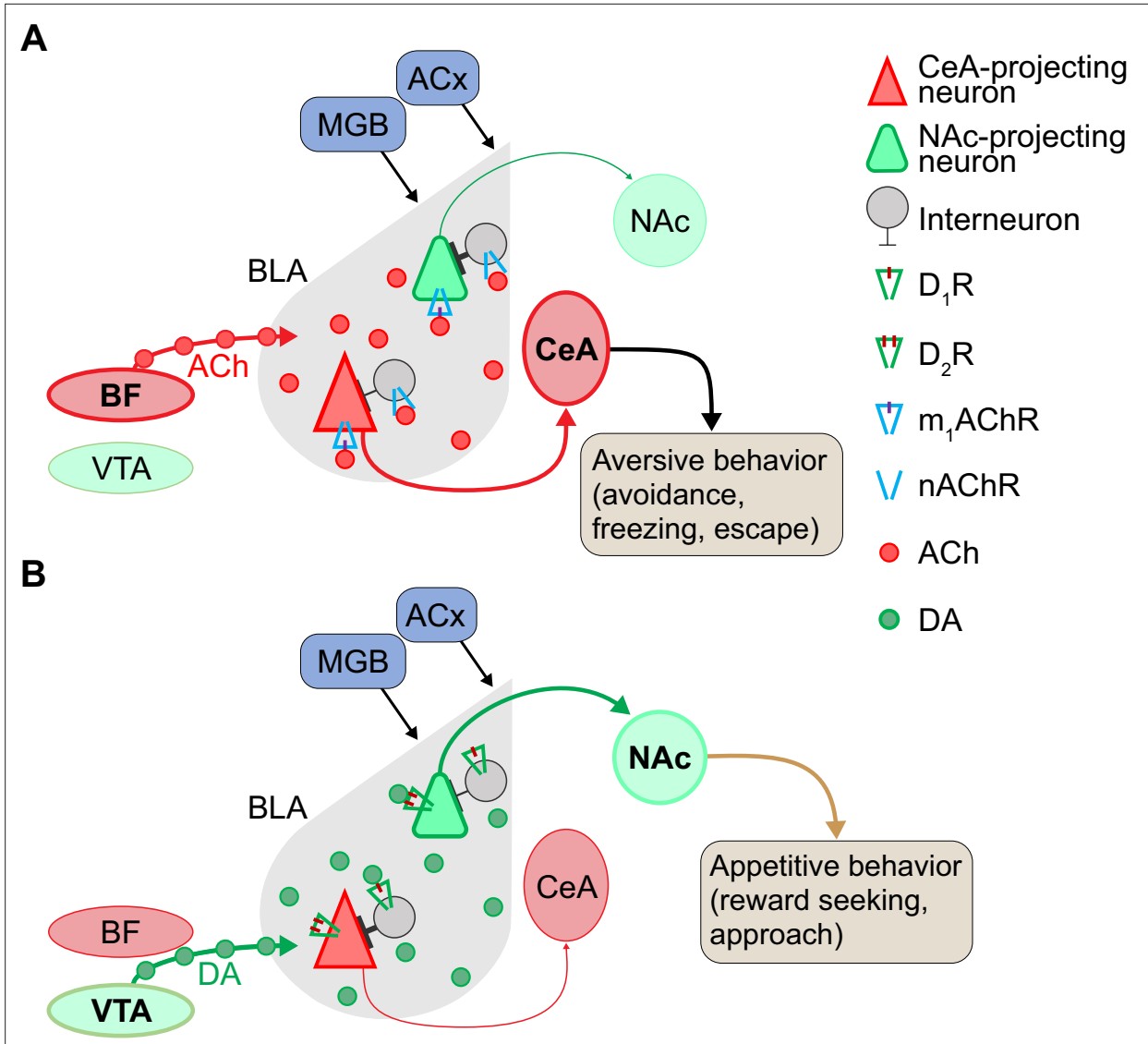

**Figure 7.** Proposed model for neuromodulation of salient vocalization processing via acetylcholine (ACh) and dopamine (DA) in the basolateral amygdala (BLA). (**A**) Cholinergic modulation of CeA-projecting neurons during aversive vocalization cue processing in the BLA. In the presence of aversive cues, ACh released from the basal forebrain acts on M1 ACh receptors (M1 mAChRs) to enhance the cue-induced excitatory responses of CeA-projecting neurons. In contrast, NAc-projecting neurons are quiescent, because they do not respond to aversive cues and are inhibited by interneurons that are activated through nicotinic ACh receptors (nAChRs). (**B**) Dopaminergic modulation and enhancement of signal-to-noise ratio in response to reward-associated cues (appetitive vocalizations). When vocalizations or other rewarding cues are present, release of DA from VTA enhances D2R-mediated excitation in NAc-projecting neurons that are responsive to positive cues. In contrast, CeA-projecting neurons are not responsive to rewarding vocalizations and are inhibited by local interneurons. DA is thought to act on D1 DA receptors (D1Rs) in these local interneurons to shape a direct inhibition onto CeA-projecting neurons. Abbreviations: ACx, auditory cortex; BF, basal forebrain; CeA, central nucleus of amygdala; MGB, medial geniculate body; NAc, nucleus accumbens; VTA, ventral tegmental area.

can result in long afterdepolarizations that produce persistent firing lasting as long as ACh is present (*Jiang et al., 2016*; *Unal et al., 2015*). Such a process may explain persistent firing observed in single neuron responses to aversive social vocalizations in bats (*Gadziola et al., 2012*; *Peterson and Wenstrup, 2012*). Through this process of inhibiting quiescent neurons and enhancing activation and persistent firing in active neurons, ACh sharpens the population signal-to-noise ratio (SNR) during the processing of salient, aversive signals in the BLA. These neurons, processing negative cues, likely project to the central nucleus of the amygdala (CeA) to regulate defensive behaviors such as escape and avoidance (*Figure 7A*; *Beyeler et al., 2016*; *Davis et al., 2010*; *Namburi et al., 2015*). In agreement, our behavioral findings show increased behaviors such as flinching along with increased release

of ACh during processing of aversive vocalizations. Such prolonged afterdepolarizations provide the appropriate condition for associative synaptic plasticity (*Likhtik and Johansen, 2019*) and underlie an increase in the ratio of alpha-amino-3-hydroxy-5-methyl-4-isoxazole propionic acid (AMPA) receptor current to N-methyl-D-aspartate (NMDA) receptor current among CeA-projecting neurons during processing of aversive cues.

Dopaminergic innervation from the ventral tegmental area (*Asan, 1998*) acts on BLA neurons via D1 and D2 receptors, both G-protein-coupled receptors. DA is important in reward processing, fear extinction, decision making, and motor control (*Ambroggi et al., 2008*; *Di Ciano and Everitt, 2004*; *Lutas et al., 2019*). We observed increased DA release in the BLA in EXP groups in response to mating vocalizations both for males and for females across estrous stages. Electrophysiological studies show that DA enhances sensory processing in BLA neurons by increasing the population response SNR in a process like ACh (*Kröner et al., 2005*; *Vander Weele et al., 2018*). Thus, during processing of mating vocalizations or those related to other rewarding experiences, DA presence in the vicinity of BLA pyramidal neurons and interneurons is enhanced (*Figure 7B*). For neurons with elevated spiking activity during processing of appetitive vocalizations or other sensory stimuli, DA acts on D2 receptors of pyramidal cells to further enhance neuronal firing and result in persistent firing of these projection neurons. Conversely, in BLA projection neurons that do not respond to such positive cues, such as CeA-projecting neurons, DA exerts a suppressive effect directly via D1 receptors and indirectly by activating inhibitory interneuron feedforward inhibition (*Kröner et al., 2005*). The net result of this process in response to appetitive vocalizations is an enhancement of activity in the reward-responding neurons and suppression of activity in aversive-responding neurons. This process likely depends on the increase in synaptic plasticity via an enhanced ratio of AMPA receptor current to NMDA receptor current during processing such cues in NAc-projecting neurons in the BLA (*Namburi et al., 2015*; *Otani et al., 2003*; *van Vugt et al., 2020*). Our findings suggest that this may occur in the BLA in response to appetitive vocalizations.

As the results with males listening to restraint vocalizations demonstrate, increased ACh release in BLA is associated with processing aversive cues. How, then, should the increased ACh release in estrus females during mating vocal playback be interpreted? Previous work shows that neuromodulation of amygdalar and other forebrain activity is altered by sex hormone/receptor changes in males and females (*Egozi et al., 1986*; *Kalinowski et al., 2023*; *Kirry et al., 2019*; *Matsuda et al., 2002*; *Mizuno et al., 2022*; *van Huizen et al., 1994*). For instance, the cholinergic neurons that project to the BLA, originating in the basal forebrain, exhibit high expression of estrogen receptors that is influenced by a female's hormonal state (*Shughrue et al., 2000*). During estrus, the enhanced circulating estrogen affects release of ACh and may influence neuronal networks and behavioral phenotypes in a distinct manner (*Gibbs, 1996*; *McEwen, 1998*). Thus, increased ACh release in estrus females may underlie increased attentional and risk-assessment behaviors in response to vocalization playback, as we observed with flinching behavior in this experimental group. Combined with DA increase, it may trigger both NAc and CeA circuit activation, resulting in both reward-seeking and cautionary behaviors in estrus females.

Our results demonstrate the strong impact of even limited experience in shaping behavioral and neuromodulatory responses associated with salient social vocalizations. In the adult mice, a single 90-min session of mating and of restraint, occurring 4–5 days prior to the playback experiment, resulted in consistent behavioral responses and consistent and enhanced ACh release into BLA for both vocalization types. As previous work shows (*Huang et al., 2012*; *Nadim and Bucher, 2014*; *Pawlak et al., 2010*), neuromodulatory inputs play crucial roles in regulating experience-dependent changes in the brain. However, it remains unclear whether the experience shapes neuromodulator release, or whether neuromodulators deliver the experience-related effect into the BLA. It is also unknown whether these experiences were specific to the type of vocalization playback associated with the experience, or whether either experience had a more generalized effect on responses to all vocalization playback, as suggested by the broad physiological effects of mating or restraint (*Leuner et al., 2010*; *Arnold et al., 2019*).

The interaction between ACh and DA is thought to shape motor responses to external stimuli (*Lester et al., 2010*). Our results support the view that a balance of DA and ACh may regulate the proper behavioral response to appetitive and aversive auditory cues. For instance, increased reward-seeking behavior (rearing and locomotion) in EXP males during mating vocalizations playback may

result from the differential release of the two neuromodulators—decreased ACh and increased DA. Furthermore, the lack of this differential release may be the underlying cause for the lack of such responses in INEXP male mice. This supports the role of experience in tuning interactions of these two neuromodulators throughout the BLA, for shaping appropriate behaviors. Overall, the behavioral changes orchestrated by the BLA in response to emotionally salient stimuli are most likely the result of the interaction between previous emotional experiences, hormonal state, content of sensory stimuli, and sex of the listening animals.

## Materials and methods

### Animals

Experimental procedures were approved by the Institutional Animal Care and Use Committee at Northeast Ohio Medical University (protocol 18-09-207). A total of 83 adult CBA/CaJ mice (Jackson Labs, p90-p180), male and female, were used for this study. Animals were maintained on a reversed dark/light cycle and experiments were performed during the dark cycle. The mice were housed in same-sex groups until the week of the experiments, during which they were singly housed. Food and water were provided ad libitum except during the experiment.

The estrous stage of female mice was evaluated based on vaginal smear samples obtained by sterile vaginal lavage. Samples were collected using glass pipettes filled with double distilled water, placed on a slide, stained using crystal violet, and coverslipped for microscopic examination. Estrous stage was determined by the predominant cell type: squamous epithelial cells (estrus), nucleated cornified cells (proestrus), or leukocytes (diestrus) (*McLean et al., 2012*). To confirm that the stage of estrous did not change during the experiment day, samples obtained prior to and after data collection on the experimental day were compared.

### Experimental overview

The basic experimental structure that occurred over 6 days is illustrated in *Figure 1D*. Briefly, each mouse was placed in an arena on Days 1 and 2 to provide one-time experiences of mating and sustained restraint. After the experience on Day 2, the subject was anesthetized for implantation of a guide tube for the microdialysis probe. On Day 6, the vocalization playback session occurred. The mouse was briefly anesthetized for insertion of the microdialysis probe into the amygdala, followed by several hours recovery. Before, during, and after vocal playback, we sampled extracellular fluid from the amygdala through the microdialysis probe and recorded video to analyze the subject's behavior. An LC/MS method was used to measure concentrations of several neurochemicals.

The microdialysis sampling interval of 10 min was used to establish the temporal framework for vocal playback and analysis of neurochemicals and behaviors (*Figure 1E*). The analysis compared neurochemicals and behaviors in a period immediately preceding the vocal stimuli (Pre-Stim) and in two 10-min periods featuring playback of the same sequence of vocal stimuli (Stim 1 and Stim 2). Each playback session featured only one category of vocalizations (associated with mating or restraint) and each animal participated in a single playback session. Additional details of this experimental structure are described in the sections below.

### Acoustic methods

#### Vocalization recording and analysis

To record vocalizations for use in playback experiments, mice were placed in an open-topped plexiglass chamber (width, 28 cm; length, 28 cm; height, 20 cm), housed within a darkened, single-walled acoustic chamber (Industrial Acoustics, New York, NY) lined with anechoic foam (*Grimsley et al., 2016*; *Niemczura et al., 2020*). Acoustic signals were recorded using ultrasonic condenser microphones (CM16/CMPA, Avisoft Bioacoustics, Berlin, Germany) connected to a multichannel amplifier and A/D converter (UltraSoundGate 416H, Avisoft Bioacoustics). The gain of each microphone was independently adjusted once per recording session to optimize the SNR while avoiding signal clipping. Acoustic signals were digitized at 500 kHz and 16-bit depth, monitored in real time with RECORDER software (Version 5.1, Avisoft Bioacoustics), and Fast Fourier Transformed (FFT) at a resolution of 512 Hz. A night vision camera (VideoSecu Infrared CCTV), centered 50 cm above the floor of

the test box, recorded the behaviors synchronized with the vocal recordings (VideoBench software, DataWave Technologies, version 7).

To record mating vocalizations, 10 animals (5 male–female pairs) were used in sessions that lasted for 30 min. A male mouse was introduced first into the test box, followed by a female mouse 5 min later. Vocalizations were recorded using two ultrasonic microphones placed 30 cm above the floor of the recording box and 13 cm apart. See below for analysis of behaviors during vocal recordings.

To record vocalizations during restraint, six mice (four male, two female) were briefly anesthetized with isoflurane and then placed in a restraint jacket as described previously (*Grimsley et al., 2016*). Vocalizations were recorded for 30 min while the animal was suspended in the recording box. Since these vocalizations are usually emitted at lower intensity compared to mating vocalizations, the recording microphone was positioned 2–3 cm from the snout to obtain the best SNR.

Vocal recordings were analyzed offline using Avisoft-SASLab Pro (version 5.2.12, Avisoft Bioacoustics) with a hamming window, 1024 Hz FFT size, and an overlap percentage of 98.43. For every syllable, the channel with the higher amplitude signal was extracted using a custom-written Python code (https://github.com/GavazziDA/Wenstrup_Lab_Ghasemahmad_2023, copy archived at *Gavazzi, 2024*) and analyzed. Since automatic syllable tagging did not allow distinguishing some syllable types such as noisy calls and MFVs from background noise, we manually tagged the start and end of each syllable, then examined spectrograms to measure several acoustic features and classify syllable types based on *Grimsley et al., 2011*; *Grimsley et al., 2016*.

## Vocalization playback

Vocalization playback lasting 20 min was constructed from a set of seven repeating stimulus blocks lasting 2:50 min each (*Figure 1E*). Each block was composed of a set of vocal sequence exemplars that alternated with an equal duration of background sound (Silence) associated with the preceding exemplar. The exemplars were recorded during mating interactions and restraint, and selected based on high SNR, correspondence with behavioral category by video analysis, and representation of the spectrotemporal features of vocalizations emitted during mating and restraint (*Ghasemahmad, 2020*; *Grimsley et al., 2016*). Mating stimulus blocks contained five exemplars of vocal sequences emitted during mating interactions. These exemplars ranged in duration from 15.0 to 43.6 s. Restraint stimulus blocks included seven vocal sequences, emitted by restrained male or female mice, with durations ranging from 5.7 to 42.3 s. Across exemplars, each stimulus block associated with both mating and restraint included different sets of vocal categories (*Figure 1A–C*).

Playback sequences, that is, exemplars, were conditioned in Adobe Audition CC (2018), adjusted to a 65-dB SNR level, then normalized to 1 V peak-to-peak for the highest amplitude syllable in the sequence. This maintained relative syllable emission amplitude in the sequence. For each sequence, an equal duration of background noise (i.e., no vocal or other detected sounds) from the same recording was added at the end of that sequence (*Figure 1E*). A 5-ms ramp was added at the beginning and the end of the entire sequence to avoid acoustic artifacts. A MATLAB app (EqualizIR, Sharad Shanbhag; https://github.com/TytoLogy/EqualizIR, *Shanbhag, 2018*) compensated and calibrated each vocal sequence for the frequency response of the speaker system. Vocal sequences were converted to analog signals at 500 kHz and 16-bit resolution using DataWave (DataWave SciWorks, Loveland, CO), anti-alias filtered (TDT FT6-2, fc = 125 kHz), amplified (HCA-800II, Parasound, San Francisco, CA), and sent to the speaker (LCY, K100, Ying Tai Audio Company, Hong Kong). Each sequence was presented at peak level equivalent to 85 dB SPL.

## Behavioral methods

Behaviors during both vocalization recording and playback sessions were recorded using a night vision camera (480TVL 3.6 mm, VideoSecu), centered 50 cm above the floor of the test box, and SciWorks (DataWave, VideoBench version 7) for video acquisition and analysis.

## Analysis of mating behaviors during vocal recordings

Interactions between male and female mice were video-recorded, then analyzed second-by-second. Among more general courtship interactions, we identified a set of behaviors that are mating-related, as described previously (*Gaub et al., 2016*; *Heckman et al., 2016*). All vocal sequences selected as exemplars for playback of 'mating' vocalizations were recorded in association with these male-mating

behaviors: head-sniffing, attempted mounting, or mounting. Vocalizations during these behaviors included chevron, stepped, and complex USVs emitted with longer durations and higher repetition rates, and more LFH calls (*Gaub et al., 2016*; *Ghasemahmad, 2020*; *Hanson and Hurley, 2012*).

## Experience and playback sessions

Prior to playback experiments, each animal underwent 90-min sessions on two consecutive days (Days 1 and 2) that provided both mating and restraint experiences to the EXP group (*n* = 31 animals) or no experiences of these to the INEXP group (*n* = 22 animals). EXP sessions were presented in a counterbalanced pattern across subjects (*Figure 1D*). For the mating experience, mounting or attempted mounting was required for the animal to be included in the remainder of the experiment. However, we did not record detailed behaviors or track estrous stage during the mating experience session. After the Day 2 session, mice underwent surgery for implantation of a microdialysis guide canula (see below), then recovered for 4 days.

On Day 6, the day of the playback experiment, male mice were randomly assigned to either restraint or mating vocal playback groups. Females were only tested with mating playback, since a preliminary behavioral study suggested no difference in male and female responses to restraint. Video recording was performed simultaneously with microdialysis, beginning 10 min before vocal playback (pre-stimulus) and continuing for 20 min during playback (*Figure 1D*).

## Analysis of behaviors during vocal playback

Behavioral analysis was based on previous descriptions (*Bakshi and Kelley, 1993*; *Bakshi and Kelley, 1994*; *Blanchard et al., 2003*; *Füzesi et al., 2016*; *Grimsley et al., 2015*; *Lezak et al., 2017*; *Saibaba et al., 1996*) (Mouse Ethogram: https://mousebehavior.org). The list of behaviors assessed was determined through pilot analysis and previous studies characterizing rodent defense and fear behaviors. They are defined as the following: abrupt attending, flinching, locomotion, rearing, grooming, still-and-alert, and stretch-attend (description in *Table 1*; *Blanchard et al., 2003*). A behavior was only counted when its occurrence lasted two or more seconds, except for flinching, which could take place more quickly.

For analysis of behaviors during vocal playback, all videos were examined blind to the sex, estrous state, or experience of the animal and the context of vocalizations. Video recordings were analyzed manually (*n* = 48 mice) in 10-s intervals for the seven behaviors identified above. The number of occurrences of every behavior per 10-s block was marked. The numbers were added for every block of 2:50 min of vocal playback (see *Figure 1E*), then averaged for the Pre-Stim, Stim 1, and Stim 2 periods separately.

Videos were further analyzed automatically using the video tracker within VideoBench (DataWave Technologies, version 7) for total distance traveled and time spent in the periphery and center of the test arena. For video-tracking analysis of time spent, the floor was divided into 16 equal squares with 4 central squares identified as 'center' and the 4 corner squares identified as periphery. The two squares in the middle of each wall were excluded from the analysis to avoid attraction to the holes affect the animal's preference for the periphery vs center.

## Procedures related to microdialysis

### Surgery

Since both left and right amygdala are responsive to vocal stimuli in human and experimental animal studies (*Wenstrup et al., 2020*), we implanted microdialysis probes into the left amygdala to maintain consistency with other studies in our laboratory. Mice were anesthetized with Isoflurane (2–4%, Abbott Laboratories, North Chicago, IL) and hair overlying the skull was removed using depilatory lotion. A midline incision was made, then the skin was moved laterally to expose the skull. A craniotomy ($\approx$1 mm$^2$) was made above the BLA on the left side (stereotaxic coordinates from bregma: $-1.65$ mm rostrocaudal, $+3.43$ mm mediolateral). A guide cannula (CMA-7, CMA Microdialysis, Sweden) was implanted to a depth of 2.6 mm below the cortical surface and above the left BLA (*Figure 1D*, Day 2), then secured using dental cement. After surgery, the animal received a subcutaneous injection of carprofen (4 mg/kg, s.c.) and topical anesthetic (lidocaine) and antibiotic cream (Neosporin). It was returned to its cage and placed on a heating pad until fully recovered from anesthesia. Recovery from the surgical procedures occurred in the animal's home cage and was assessed by resumption of

normal eating, drinking, exploratory, and grooming behaviors and postures. Playback experiments occurred in all recovered animals 4 days after surgery.

## Microdialysis

On the day before microdialysis, the probe was conditioned in 70% methanol and artificial cerebrospinal fluid (aCSF) (CMA Microdialysis, Sweden). On playback day (Day 6), the animal was briefly anesthetized and the probe with 1-mm membrane length and 0.24-mm outer diameter (MWCO 6 kDa) was inserted into the guide cannula (*Figure 1D*).

Using a spiral tubing connector (0.1 mm ID × 50 cm length) (CT-20, AMUZA Microdialysis, Japan), the inlet and outlet tubing of the probe was connected to the inlet/outlet Teflon tubing of the microdialysis lines. A swivel device for fluids (TCS-2-23, AMUZA Microdialysis, Japan), secured to a balance arm, held the tubing, and facilitated the animal's free movement during the experiment.

After probe insertion, a 4-hr period allowed animals to habituate and the neurochemicals to equilibrate between aCSF fluid in the probe and brain extracellular fluid. Sample collection then began, in 10-min intervals, beginning with four background samples, then two samples during playback of restraint or mating vocal sequences (total 20 min), then one or more samples after playback ended. To account for the dead volume of the outlet tubing, a flow rate of 1.069 µl/min was established at the syringe pump to obtain a 1-µl sample per minute. Samples collected during this time were measured for volume to assure a consistent flow rate. To prevent degradation of collected neurochemicals, the outlet tubing was passed through ice to a site outside the sound-proof booth where samples were collected on ice, then stored in a −80°C freezer. The choice of 10-min sampling intervals was based on the minimum collection time needed to provide 5 µl samples both for immediate testing and for a backup sample in the event of catastrophic failure of the processes associated with the neurochemical analysis.

## Neurochemical analysis

Samples were analyzed using an LC/MS technique (*Figure 1D*) at the Vanderbilt University Neurochemistry Core. This analysis was blind to the sex, estrous state, or experience of the animal and the context of vocalizations. This method allows simultaneous measurement of several neurochemicals in the same dialysate sample: ACh, DA, and its metabolites (3,4-dihydroxyphenylacetic acid (DOPAC) and homovanillic acid (HVA)), serotonin (5-HT) and its metabolite (5-HIAA), norepinephrine (NE), gamma aminobutyric acid (GABA), and glutamate. Due to low recovery of NE and 5-HT from the mouse brain, we were unable to track these two neuromodulators in this experiment.

Before each LC/MS analysis, 5 µl of the sample was derivatized using sodium carbonate, benzoyl chloride in acetonitrile, and internal standard (*Wong et al., 2016*). LC was performed on a 2.0 × 50 mm, 1.7 µM particle Acquity BEH C18 column (Waters Corporation, Milford, MA, USA) with 1.5% aqueous formic acid as mobile phase A and acetonitrile as mobile phase B. Using a Waters Acquity Classic UPLC, samples were separated by a gradient of 98–5% of mobile phase A over 11 min at a flow rate of 0.6 ml/min with delivery to a SCIEX 6500+Qtrap mass spectrometer (AB Sciex LLC, Framingham, MA, USA). The mass spectrometer was operated using electrospray ionization in the positive ion mode. The capillary voltage and temperature were 4 kV and 350°C, respectively (*Wong et al., 2016*; *Yohn et al., 2020*). Chromatograms were analyzed using MultiQuant 3.0.2 Software (AB SCIEX, Concord, Ontario, Canada).

## Verification of probe location

We verified placement of microdialysis probes to minimize variability that could arise due to placement in regions surrounding BLA that receive different sources of neurochemical inputs (e.g., cholinergic inputs to putamen and central amygdala). To verify probe placement within the BLA, each probe was perfused with 2% dextran-fluorescein (MW 4 kDa) (Sigma-Aldrich Inc, Atlanta, GA) at the end of the experiment. The location of the probe was then visualized in adjacent cleared and Nissl-stained sections (e.g., *Figure 2*, insets). Sections were photographed using a SPOT RT3 camera and SPOT Advanced Plus imaging software (version 4.7) mounted on a Zeiss Axio Imager M2 fluorescence microscope. Adobe Photoshop CS3 was used to adjust brightness and contrast globally. Animals were included in statistical analyses only if ≥75% of the probe membrane was located within the BLA (*Figure 2*).

## Data analysis

For neurochemical analyses, the total numbers of animals used were $N_{EXP}$ = 31 and $N_{INEXP}$ = 22. Only mice with usable neurochemical data were further evaluated for behaviors. This included 46 mice: ($N_{EXP}$ = 25 and $N_{INEXP}$ = 21). Note that some animals used in neurochemical analyses were not included in the behavioral analyses because their behavioral data were unavailable. Since only one INEXP female was in a non-estrus stage during the playback session, our analysis of the effect of experience included only estrus females and males. Furthermore, one INEXP female without detectable DA levels was removed from the INEXP group neurochemical analysis.

Only the following 10-min sample collection windows were used for statistical analysis: Pre-Stim, Stim 1, and Stim 2 (*Figure 1E*). The purpose of the two collection windows during vocal playback was to allow detection of different epochs of neurochemical release, as may occur in ACh release into the amygdala (*Aitta-Aho et al., 2018*). All neurochemical data were normalized to the background level, obtained from a single pre-stimulus sample immediately preceding playback. This provided clarity in representations and did not result in different outcomes of statistical tests compared to use of three pre-stimulus samples. The fluctuation in all background samples was ≤20%. The percentage change from background level was calculated based on the formula: % change from background = (100 × stimulus sample concentration in pg)/background sample concentration in pg. Values are represented as mean ± one standard error unless stated otherwise. Raw neurochemical values are reported in *Figure 3—source data 1–3* for ACh,DA, and 5-HIAA, respectively.

All statistical analyses were performed using SPSS (IBM, V. 26 and 27). To examine vocalizations in different stages of mating, we used a linear mixed model to analyze the changes of interval and duration (dependent variables) of vocalizations based on mating interaction intensity (fixed effect). For behavioral and neurochemical analyses in playback experiments, we initially compared the output using a linear mixed model and a generalized linear model (GLM) with a repeated measure for neurochemical data. Since both statistical methods resulted in similar findings, we chose to use the GLM for statistical comparisons of the neurochemicals and behaviors. Where Mauchly's test indicated that the assumption of sphericity had been violated, degrees of freedom were corrected using Greenhouse–Geisser estimates of sphericity.

To further clarify the differences observed between groups for every comparison in the GLM repeated measure, multivariate contrast was performed. All multiple comparisons were corrected using Bonferroni post hoc testing. 95% confidence intervals were used to compare values between timepoints (*Julious, 2004*).

For both microdialysis and behavioral data, we tested the hypotheses that the release of neurochemicals into the BLA and the number of behaviors is differently modulated by the vocal playback type (mating and restraint) in male mice, by estrous stage of females or sex of the animals during mating vocal playback, and by previous experience in any of these groups. In the GLM model for *Figure 3*, context of vocalizations was defined as the fixed factor and time (Stim 1 and Stim 2) as the within-subject factor. The model in *Figure 4* tested the effect of sex and estrous state as a nested variable within sex as fixed factors, which were considered in the model design (design = intercept + sex + estrous (sex)). The GLM model in *Figure 5* tested sex, context, and experience as fixed factors with the interaction terms of sex*experience and context*experience. Here, we avoided using a full factorial model due to the absence of females in the restraint group, as that would have made the model unbalanced (design = intercept + sex + context + experience + sex*experience + context*experience). Furthermore, estrous state was not tested in examining the experience effect due to lack of INEXP non-estrus females. Dependent (response) variables included the normalized concentration of ACh, DA, 5-HIAA in Stim 1 and Stim 2 or the numbers of behaviors in Pre-Stim, Stim 1, and Stim 2 as previously described in the behavioral analysis section.

Throughout the figures, the distributions of data points are visualized by box plots, computed using the 'inclusive median' formula in Excel. The box indicates first (Q1) and third (Q3) quartiles with the median line between the two. The whiskers normally stretch between minimum and maximum values, but this computation sometimes shows data points outside of the whiskers. Excel identifies these values as outliers in the boxplot function. However, our GLM statistical analysis includes these values because there was no compelling reason to exclude them. The mean value is marked with an 'x'.

## Acknowledgements

We are indebted to Sharad Shanbhag and Daniel Gavazzi for software used to condition vocal sequences and analyze audio and behavioral data, respectively, to Kristin Yeager of the Kent State University Statistical Consulting Center for statistical consulting, and to Sheila Fleming for advice on behavioral assessments. We thank Anthony Zampino, Austin Poth, Debin Lei, and Krish Nair for technical assistance. We further thank Vanderbilt University Neurochemistry Core, supported by Vanderbilt Brain Institute and the Vanderbilt Kennedy Center, that performed the neurotransmitter sample analyses. We are grateful to Drs. Alexander Galazyuk, Brett Schofield, Merri Rosen, Mahtab Tehrani, and Sharad Shanbhag for their comments on previous versions of this manuscript. National Institutes of Health grant R01DC00937 (JJW, Alexander Galazyuk). Kent State University Graduate Student Senate (ZG). Northeast Ohio Medical University Biomedical Sciences Program Committee (ZG).

## Additional information

### Funding

| Funder | Grant reference number | Author |
|---|---|---|
| National Institute on Deafness and Other Communication Disorders | R01DC00937 | Jeffrey J Wenstrup |
| Kent State University | Graduate Student Senate Research Award | Zahra Ghasemahmad |
| Northeast Ohio Medical University | BMS Dissertation Completion Award | Zahra Ghasemahmad |

The funders had no role in study design, data collection, and interpretation, or the decision to submit the work for publication.

### Author contributions

Zahra Ghasemahmad, Conceptualization, Resources, Data curation, Formal analysis, Funding acquisition, Validation, Investigation, Visualization, Methodology, Writing – original draft, Project administration, Writing – review and editing; Aaron Mrvelj, Formal analysis, Visualization; Rishitha Panditi, Bhavya Sharma, Formal analysis; Karthic Drishna Perumal, Formal analysis, Data curation; Jeffrey J Wenstrup, Conceptualization, Resources, Data curation, Supervision, Funding acquisition, Investigation, Visualization, Methodology, Writing – original draft, Project administration, Writing – review and editing

### Author ORCIDs

Zahra Ghasemahmad http://orcid.org/0000-0002-4916-3615
Karthic Drishna Perumal http://orcid.org/0000-0002-0821-9758
Jeffrey J Wenstrup https://orcid.org/0000-0002-3566-2480

### Ethics

This study was performed in accordance with recommendations in the Guide for the Care and Use of Laboratory Animals of the National Institutes of Health. All procedures utilizing animals conformed to those approved by the Institutional Animal Care and Use Committee of the Northeast Ohio Medical University (#15-033 and #18-09-207). All surgery was performed using isoflurane anesthetic. Our procedures and practice sought to minimize animal suffering.

Reviewer #1 (Public review): https://doi.org/10.7554/eLife.88838.4.sa1
Reviewer #3 (Public review): https://doi.org/10.7554/eLife.88838.4.sa2
Author response https://doi.org/10.7554/eLife.88838.4.sa3

## Additional files

### Supplementary files
• MDAR checklist

### Data availability
Numerical data used to generate the figures are contained in source data files provided for Figures 1 and 3. Software tools used in preparing sounds or analyzing sounds/behaviors are identified in the Materials and methods and available on provided links to GitHub.

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
