## [Editor Report · eLife assessment]

This **important** study advances our understanding of how distinct types of communication signals differentially affect mouse behaviors and amygdala cholinergic/dopaminergic neuromodulation. The evidence supporting the authors' claims is **solid**. Researchers interested in the complex interaction between prior experience, sex, behavior, hormonal status, and neuromodulation should benefit from this study.

---

## [Referee Report · Reviewer #1 (Public review)]

The manuscript addresses a fundamental question about how different types of communication signals differentially affect brain state and neurochemistry. In addition, their manuscript highlights the various processes that modulate brain responses to communication signals, including prior experience, sex, and hormonal status. Overall, the manuscript is well-written and the research is appropriately contextualized.

---

## [Referee Report · Reviewer #3 (Public review)]

The work by Ghasemahmad et al. has the potential to significantly advance our understanding of how neuromodulators provide internal-state signals to the basolateral amygdala (BLA) while an animal listens to social vocalizations.

Ghasemahmad et al. made changes to the manuscript that have significantly improved the work. In particular, the transparency in showing the underlying levels of Ach, DA, and 5HIAA is excellent. My previous concerns have been adequately addressed.

---

## [Author Response]

The following is the authors’ response to the previous reviews.

**Public Reviews:**

**Reviewer #1 (Public Review):**
The manuscript addresses a fundamental question about how different types of communication signals differentially affect brain state and neurochemistry. In addition, their manuscript highlights the various processes that modulate brain responses to communication signals, including prior experience, sex, and hormonal status. Overall, the manuscript is well-written and the research is appropriately contextualized.That being said, it remains important for the authors to think more about their analytical approaches. In particular, the effect of normalization and the explicit outlining and interpretations of statistical models. As mentioned in the original review, the normalization of neurochemical data seems unnecessary given the repeated-measures design of their analysis and by normalizing all data to the baseline data and including this baseline data in the repeated measures analysis, one artificially creates a baseline period with minimal variation that dramatically differs in variance from other periods (akin to heteroscedasticity). If the authors want to analyze how a stimulus changes neurochemical concentrations, they could analyze the raw data but depict normalized data in their figures (similar to other papers). Or they could analyze group differences in the normalized data of the two stimulus periods (i.e., excluding the baseline period used for normalization).

We appreciate the reviewer’s point on the difference in variance caused by including the 100% baseline values in the analysis. After consulting with our statistician, we chose the latter of the two approaches suggested by the reviewer. Specifically, we reran the analysis to exclude the baseline and focus only on the playback windows and the group differences. The text in the results, the significance signs in the figures, and the discussion are corrected accordingly. Despite these changes, our major conclusions remains as before.

We also followed this reviewer’s suggestions to clarify the statistical model in studying the experience effect. After further consultation with our statistician, we reran the analysis on experience effect, including all the groups of EXP and INEXP animals together. We have corrected text in the figure captions, results, discussion, and data analysis sections of the manuscript related to the effect of experience and its interactions. This has not changed the conclusion made related to the experience effect in the dataset.

It would also be useful for the authors to provide further discussion of the potential contributions of different types of experiences (mating vs. restraint) to the change in behavior and neurochemical responses to the vocalization playbacks and to try to disentangle sensory and motor contributions to neurochemical changes.

We have acknowledged in the Discussion that previous studies suggest that the effect of experience involving stress could be generalized. We believe that this is an important area of future research. Our Discussion acknowledges that the relationship between sensory and motor contributions to neurochemical changes remains an area of interest. We further point out that the time resolution of microdialysis data renders the suggested discussion highly speculative. We plan to use other methods to assess this in future experiments.

**Reviewer #3 (Public Review):**
The work by Ghasemahmad et al. has the potential to significantly advance our understanding of how neuromodulators provide internal-state signals to the basolateral amygdala (BLA) while an animal listens to social vocalizations.Ghasemahmad et al. made changes to the manuscript that have significantly improved the work. In particular, the transparency in showing the underlying levels of Ach, DA, and 5HIAA is excellent. My previous concerns have been adequately addressed.
**Recommendations for the authors:**

**Reviewer #1 (Recommendations For The Authors):**
I appreciate the authors responses to my previous queries (and to the comments by other reviewers). The introduction does a better job contextualizing the data, and the additional details in the results and Methods sections help readers digest the material. I continue to think the topic is interesting and the manuscript is potentially impactful. However, I continue to be concerned about their analytical approaches and other aspects of the revised manuscript.(a) NormalizationIn my original review I wrote: "The normalization of neurochemical data seems unnecessary given the repeated-measures design of their analysis and could be problematic; by normalizing all data to the baseline data (p. 24), one artificially creates a baseline period with minimal variation (all are "0"; Figures 2, 3 & 5) that could inflate statistical power." I continue to feel that an analysis of normalized data that includes the baseline data is inappropriate because of the minimal variation in the normalized data for the baseline period. When the normalized data for the baseline period is included in the analysis, there is clearly variation in the extent of variability within each of the time periods (no variability at baseline, variability during periods 1 & 2; analogous to heteroscedasticity). For example, when analyzing the RAW DATA about the change in ACh release in experienced males listening to restraint vocalizations (thank you for releasing the raw data), there was a non-significant effect of time (baseline, period 1, and period 2; linear mixed effects model; F(2,12)=3.2, p=0.0793). However, when the normalized data for this dataset was analyzed (with baseline values being set at 100% for each mouse), there was a statistically significant effect (F(2,12)=4.5, p=0.0352). This example is just to illustrate how normalization can affect (e.g., inflate) statistical power.That being said, I do think that it is reasonable to analyzed normalized data if the period used for normalization is NOT included in the analysis (see Figure 3 of one of the paper the authors listed in their response to reviewers: Galvez-Marquez et al., 2022). However, from the reading of this manuscript, it does seem like normalized baseline data are analyzed to assess how stimuli affect neurochemical concentrations.

We appreciate the reviewer’s point on the difference in variance caused by including the 100% baseline values in the analysis. After consulting with our statistician, we chose one of the two approaches suggested by the reviewer. Specifically, we reran the analysis to exclude the baseline and focus only on the playback windows and the group differences. The text in the results, the significance signs in the figures, and the discussion are corrected accordingly. Despite these changes, our major conclusions remains as before. We have included some descriptive statistics in the text because we think these are informative.

We decided to take this approach because the inter-individual variability in the raw data levels, caused by non-experimental factors, is too great to be useful. As we have stated before, these values are affected by probe placement, collection process, or differences in the HPLC or LC/MS runs. These effects are widely recognized in the field.

It is worth pointing out a few things about the papers listed by the authors. Li et al. (2023) does depict normalized microanalysis data but it isn't clear that any analysis of the normalized data is conducted. The same can be said about Holly et al. (2016). Further, in Bagley et al (2011), the authors depict normalized data in the figures but conduct analyses on the raw data ("After chronic morphine treatment, systemic naloxone injection increased GABA outflow in PAG by 41% from 24.6 ± 2.9 nM to a peak of 34.8 ± 3.8 nM, n = 6, P = 0.016), but did not alter GABA levels after vehicle treatment (39.8 ± 8.3 to 38.6 ± 7.4 nM with naloxone at matched peak time, n = 4; Fig. 3a)". This latter approach (analyzing raw data in a repeated-measures manner and depicted normalized data) seems reasonable for the authors of the current study.(b) Clarification and modification of statistical modelsWhen analyzing the effect of experience on neuromodulator release, the authors analyze the experienced and inexperienced mice independently (e.g., figure 3 vs. 6). The ideal way to assess the effects of experience is to create a factorial model. For example, one could analyze a full factorial model with experience (exp vs. inexp), stimulus time (mating vs. restraint) and time (baseline, period 1 vs period 2, assuming raw data are used). If one wanted to exclude the baseline period because group differences in baseline are not informative, conducting a factorial analysis of normalized data with just the data from period 1 and 2 seems fine. I believe an analysis like this will help increase the legitimacy of the analysis. For example, when analyzing the normalized data (periods 1 and 2) of experienced and inexperienced males in response to mating or restraint vocalizations, you find a significant interaction between experience and stimulus type. Finding an effect of experience in an analysis that includes both experienced and inexperienced mice is ideal from an analytical framework.In Figure 6, it is not clear what the statistical model is and what the interactions mean. For example, in the figure legend for figure 6, the authors report time*context and time*sex interactions. However, in this analysis there are two groups of inexperienced males (males that are listening to restraint vocalizations, males that are listening to mating vocalizations) and one group of females (females that are listening to mating vocalizations); in other words, this is an unbalanced analysis. So, when the authors indicate a time*context interaction, does that mean they are comparing the male-restraint group to the combination of males and females listening to mating vocalizations? And when they talk about a time*sex interaction, are they analyzing how males listening to either mating or restraint vocalizations differ from females listening to a mating vocalization? This all seems peculiar to me.- A similar set of questions could be raised about interaction effects depicted in Figure 4.Overall, I would like this manuscript to be reviewed by a statistician to provide additional input on how best to analyze the data.

We followed the reviewer’s suggestions to clarify the statistical model in studying the experience effect. After further consultation with the statistician, we reran the analysis on experience effect, including all the groups of EXP and INEXP animals together.

Design: Intercept + Sex +Context + Experience+ Sex* Experience + Context* Experience.

The model is not full factorial as recommended by the statistician, because we don’t have females in the restraint group and that would make an unbalanced design. Therefore, running GLM based on the above model and included factors, as advised by the statistician, is the best way of approaching the analysis for the current dataset.

We have corrected text in the figure captions, results, discussion, and data analysis sections of the manuscript related to the effect of experience and its interactions. The GLM models are clarified for all the figures in the “data analysis” section of the manuscript. We have clarified that the major effect of experience on neuromodulators was seen in the ACh data.

(c) Analysis of post-stimulus periodI agree with Reviewer 3 that analyzing the post-stimulus period would be useful. As mentioned in the original review, these data could serve as an opportunity to show that the neurochemical levels returned to baseline and add further support for the model described in Figure 6. In addition, these data could help reveal the link between neurochemical release, auditory responses, and behavior. If neurochemical changes reflect auditory responses, then these should back to baseline during the post-stimulus period. In addition, if behavioral variation (e.g., between mice hearing mating vs. restraint stimuli) persists following the termination of playback, then one could similarly assess whether neurochemical variation persists following playback. If the latter is the case, then the neurochemical release could be more related to the behavior than to the playback stimulus itself.

We did not change this analysis. Our response to Reviewer 3’s comment is shown below.

“We decided not to include analyses of the post-stimulus period because this period is subject to wider individual and neuromodulator-specific effects and because it weakens statistical power in addressing the core question—the change in neuromodulator release DURING vocal playback. We agree that the general question is of interest to the field, but we don’t think our study is best designed to answer that question.”

This was accepted by Reviewer 3. We also note that release patterns have multiple time courses (e.g., Aitta-aho et al., 2018 for ACh), and thus may not support an assumption that levels should return to baseline shortly after playback offset.

Minor comments:Page 7, line 15: I suggest changing "vocalization-dependent" to "stimulus-dependent" because the former could connote patterns of release related to the animal itself vocalizing.

Changed to: “There were also distinct patterns of ACh and DA release into the BLA depending on the type of vocalization playback (Fig 3C,D).”

Discussion section: The authors should point out a few caveats with their experiments in the Discussion section. First, experienced animals received both mating (social) and restraint experiences, and it is not clear to what degree each type of experience affected neural and behavioral responses (i.e., specificity of experience effects). For example, mating experience can lead to a wide range of physiological changes, including a resilience to stress (e.g., Leuner et al., PLoS One, 2010; Arnold et al., Hormones and Behavior, 2019), so it is possible that mating experiences by themselves could have induced these changes. Or it could be that experiencing restraint stress affects responses to mating stimuli. This could be added to the first paragraph in page 16. (The authors could also discuss which aspects of the sexual encounters might be most important for the behavioral and neural plasticity.)

We have added text to raise this issue, stating that it is unknown wither the experience effects are specific and citing the above references concerning the generalized effects of certain experiences.

Discussion section: It would also be useful for the authors to discuss the extent to which behavior might be driving the neurochemical changes. Some of the analyses suggest that the release is independent of the behavior (e.g., reflects a sensory responses) but this could be emphasized more in the Discussion.

We believe that we have addressed this issue sufficiently in our previous response to related issues raised by this reviewer. As we note, there are limitations in the time resolution of microdialysis data that render the suggested discussion highly speculative. We plan to use other methods to assess this in future experiments.

Figure 2, legend: Please note that the text above the images describes the stimulus played back to these animals and their hormonal state, and not the type of experienced they underwent (i.e., clarify the titles)

Changed as requested.

I also agree with Reviewer 3 that "mating experience" is a misnomer for this manuscript. "Social experience with a female" is a more accurate descriptor. If they wanted to specifically provide mating experience, males should have only been tested with estrus (receptive females). I don't think this wording change detracts from their findings.

We have not changed this term. As noted in our previous response to Reviewer #3, we stated: “In the mating experience, mounting or attempted mounting was required for the animal to be included in subsequent testing.” Due to this requirement, the term “mating behavior” is informative and appropriate. In our view, “Social experience with a female” does not adequately describe our inclusion criterion or the experience.

**Reviewer #3 (Recommendations For The Authors):**
The work by Ghasemahmad et al. has the potential to significantly advance our understanding of how neuromodulators provide internal-state signals to the basolateral amygdala (BLA) while an animal listens to social vocalizations.Ghasemahmad et al. made changes to the manuscript that have significantly improved the work. In particular, the transparency in showing the underlying levels of Ach, DA, and 5HIAA is excellent. My previous concerns have been adequately addressed. I only have a few minor suggestions for the text and one figure.Minor suggestions:Page 2, Ln 9: add adult before male and female mice

Changed as requested

Page 4, Ln 10: add a period after Tsukano et al., 2019

Changed as requested

Page 6, Ln 9: what did you mean by "their interaction"? Being more specific, but concise, would help the readers.

We revised the wording to clarify that the neuromodulatory systems interact in the emission of positive and negative vocalizations.

Page 6, Ln 17: You mention Stim 1 and Stim 2, but the stimuli are not defined at this point. The clear explanation is provided in the following paragraph. Maybe consider switching the order and define the stimuli before you describe the liquid chromatography/mass spectrometry technique.

We have revised and merged these paragraphs so that Stim 1 and Stim 2 are defined on first use. We also revised our description of the depiction and analysis of neurochemical data.

Page 11, Ln 12: replace well-proven with well-documented

Changed as requested

Figure 2: There are two arrows pointing towards a single track. I assume one of the arrows is a duplicate. If so, delete one of the arrows. If not, please explain what the second arrow represents.

Arrow removed